# ENHANCING LANGUAGE EMERGENCE THROUGH EMPATHY

## ABSTRACT

The emergence of language in multi-agent settings is a promising research direction to ground natural language in simulated agents. If AI would be able to understand the meaning of language through its using it, it could also transfer it to other situations flexibly. That is seen as an important step towards achieving general AI. The scope of emergent communication is so far, however, still limited. It is necessary to enhance the learning possibilities for skills associated with communication to increase the emergable complexity. We took an example from human language acquisition and the importance of the empathic connection in this process. We propose an approach to introduce the notion of empathy to multi-agent deep reinforcement learning. We extend existing approaches on referential games with an auxiliary task for the speaker to predict the listener's mind change improving the learning time. Our experiments show the high potential of this architectural element by doubling the learning speed of the test setup.

## 1 INTRODUCTION

Natural language is not as rule-based as researchers in supervised language learning would prefer. There are limitless context-dependent notions to it, and flexible language use is considered as a necessary aspect of general AI. Originally, natural language emerged through a necessity to achieve successful coordination. Hence, a general AI would need to understand the functional aspects of language and learn communication through interaction (Wittgenstein, 1958; Wagner et al., 2003). These considerations led to the research field of emergent communication and the attempt to ground natural language through reinforcement learning.

Deep reinforcement learning has achieved some impressive results over the last years (Arulkumaran et al., 2017). One of its principal aspects is the ability to extract features from high dimensional input data without manual preprocessing. This capability is especially useful if the necessary representation is unknown to the designer.

Classical deep reinforcement learning approaches rely on a large number of training examples, mainly because the sparse reward hardly provides enough feedback to shape the deep layers. These deep layers are responsible for the embedding of input data into a meaningful representation. Therefore, it takes many training steps before a useful representation emerges; if it converges at all. According to the theory of the predictive mind (Hohwy, 2013), the human brain generates richer feedback through learning several unsupervised prediction tasks while training on the main task. The purpose of these predictions is to produce more and more expressive models and representations of the world.

Oh et al. (2015) achieved a far more expressive representation of their visual inputs by learning an auxiliary prediction task. The sole purpose of the auxiliary net is to predict the change in the visual input given the last movement action. Training this net does not directly affect the original task, but it refines the visual representation to reflect the concepts of a 3D world. Hermann et al. (2017) used predictive tasks to ground natural language, but only focused on better understanding an existent language. We transfer the auxiliary prediction to the task of active communication. This goes along with the theory of mind (Premack & Woodruff, 1978; Schaafsma et al., 2015) stating that an essential part of intelligence in interaction emerges through predicting the mental state of the interaction partner.

We let the speaker train an auxiliary net that tries to predict how the speaker's utterance will change the listener's hidden state. That resembles humans empathetic way of understanding what a message will do to the listener. We assume this leads to a more communication effective representation of the sensory input; in other words, the input encoding becomes more communicatable. The effect is visible in the essential acceleration of learning successes in developing a shared language.

*Our main contribution* is an elegant extension to multi-agent deep reinforcement learning (MADRL) algorithms aiming to emerge a communication. It resembles an empathic connection between speaker and listener, which leads to faster convergence to a shared language. We doubled the learning speed of a MADRL algorithm playing a referential game by introducing this auxiliary prediction task to the speaking agent. We attribute the improvement to the richer gradients in the lower layers of the neural network to embed the input.

## 2 BACKGROUND

**Reinforcement Learning (RL)** An agent in a reinforcement learning setting can fully or partially observe its current state $s \in S$ and is able to choose an action $a \in A$ through a policy $\pi(s) = a$. The chosen action will lead to receiving a reward $R$. The agent's goal in its environment is to maximize the expected reward (Sutton et al., 1998).

$$J(\theta) = \mathbb{E}[R(s, a)] \qquad (1)$$

**RL with neural networks (NN)** Using neural networks as a policy representation for reinforcement learning has the benefit of being able to represent any policy function and the downside of needing a huge number of data samples to learn. In our case, the policy outputs a direct probability for taking each action. Such policies can be updated by using Policy Gradient methods (Sutton et al., 2000). The policy parameters $\theta$, in this case, the parameters of the neural net, are updated according to their effect on the objective $J$ with a learning rate $\beta$:

$$\Delta\theta \approx \beta \frac{\delta J(\theta)}{\delta \theta} \qquad (2)$$

Using the REINFORCE algorithm (Williams, 1992) the effect on the objective can be estimated as the following:

$$\nabla_\theta J(\theta) = \nabla_\theta \log \pi(a|s) R(s, a) \qquad (3)$$

**Long Short-Term Memory Network (LSTM)** Recurrent neural networks (RNN) can accumulate input in an internal representation over time as well as produce a consistent output over several time steps from it. LSTMs are RNNS that are specifically created to remember information over an extended period of steps (Hochreiter & Schmidhuber, 1997).

**Auxiliary tasks in RL** Auxiliary unsupervised tasks were introduced into RL with NNs by Oh et al. (2015). They proposed an architecture that predicts the next visual input, given the internal representation of the last visual inputs and the last taken action. The unsupervised task of correctly predicting the next visual input leads to better performance on the main task, which was playing an atari game. They assume that the auxiliary task enforces a more expressive internal representation of the visual input, which then aids the main task. Hermann et al. (2017) transferred this auxiliary task to natural language acquisition by predicting the next word spoken by an instructor.

## 3 RELATED WORK

MacLennan (1990) started the field of learning communication in artificial agents with the aim to research the mechanisms by which language and communication emerge. Werner & Dyer (1991) contributed by using classical genetic scenarios, where "male" agents had to find "female" agents based on signals they emitted. They extended their setting in 1993 to include predator and prey agents and showed that known prey strategies as herding emerge if the agents have the possibility to communicate(Werner & Dyer, 1993). Robbins (1994) achieved the emergence of more robust signals by introducing lying agents (parasites) in this setting.
The successive advances in the field of learning communication can be assigned to the progress of learning algorithms for neural networks for a big part. Kasai et al. (2008) started using Q-learning

on the pursuit problem, including learned communication, but also follow up work only reached simple information sharing about the prey position (Kasai et al., 2009; Noro et al., 2014).

The field was then alleviated to a new level of complexity by Sukhbaatar et al. (2016) and Foerster et al. (2016). They transferred the new progress in deep learning to multi-agent coordination to emerge even more complex communication patterns. In previous work, we have shown that these algorithms can be further improved even to solve tasks that lie outside the communication range [blind].

Though many multi-agent learning setups use communication as a matter of success, only some focus on the emerging protocols and their properties (Hernandez-Leal et al., 2018). From those focusing on the emergence of communication or language, a significant number of publications used referential games (Lewis, 2008) as a testbed (Jorge et al., 2016; Lazaridou et al., 2017; Havrylov & Titov, 2017; Das et al., 2017; Evtimova et al., 2017; Lazaridou et al., 2018).

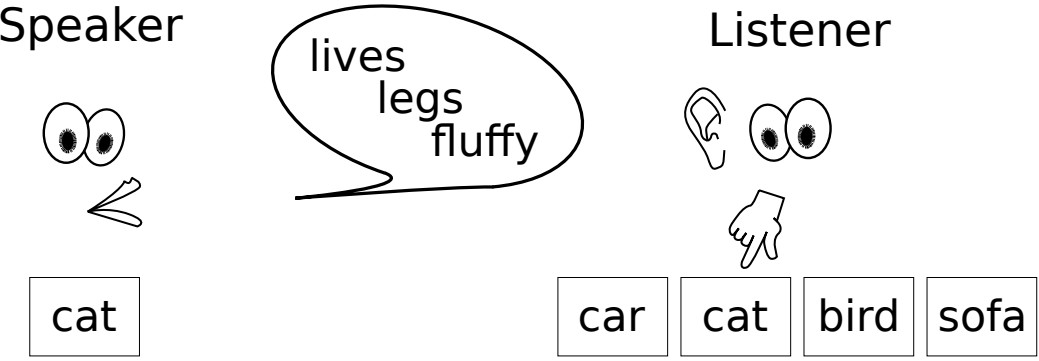

Figure 1: Illustration of the referential game used as a testbed. The speaker agent takes the concept "cat" as input and forms a message to describe it. The listener agent takes the message and several candidate concepts as input, and decides which concept is the target seen by the speaker.

Especially interesting in that context is the work of Lazaridou et al. (2017), as they could vividly show, that this approach to language emergence can lead to a flexible language use which could be understood by humans even when applied to objects unknown to the algorithm, yet.

## 4    CONTRIBUTIONS

We introduce the idea of auxiliary tasks into the field of language emergence. The speaking agent is equipped with an auxiliary single-layer perceptron, to predict the hidden state of the listener agent, after this ultimately encoded the message. The input for this prediction is the hidden state of the speaker right before it starts forming the message. The aim is to achieve a high relation between the hidden states of both agents. This signifies the speaker can communicate its means well. We state that in this application, the auxiliary prediction resembles empathy in humans, as the speaker tries to predict how its utterance will affect the listener's mindset.

The prediction task is unsupervised and can be trained on the same samples and at the same time as the main task. Training the main RL task automatically generates the samples for the unsupervised task. The gradients can be backpropagated into the encoding layers of the speaker, where they are added to the gradients of the RL task and optimized together. With our approach, we further enhance the possibilities in language emergence by providing richer feedback to form the internal communicatable representation in the speaking agent. We provide experimental evidence that these extensions can lead to a doubled learning speed when added to an existing approach to language emergence in referential games.

## 5 EXPERIMENTS

To test the potential of the auxiliary prediction task, we used the referential game setup proposed by Lazaridou et al. (2018) shown in Fig. 1. Out of the existing implementations, we chose this one because the setup has proven to converge to an emergent communication at a relatively low computational cost.

**Dataset** We use the Visual Attributes for Concepts Dataset (VisA) of Silberer et al. (2013). It contains attribute annotations for 500 concrete concepts (like cat or sofa), annotated with 636 general attributes (like is_black or made_of_wood). The annotations are human-made and therefore carry an inherent structure that can be seen as disentangled.

**Agent Setup** A speaker agent gets shown a target concept $t$ that is realized as a binary vector with as many entries as possible attributes. The speaker then uses a policy $\pi^S$ to produce a message $m$ out of an alphabet of discrete symbols (numbers 1 to 100 in our case). The message is then interpreted by a listener agent that observes several candidate concepts $C$ at the same time. The listener uses a pointing policy $\pi^L$ to decide, which of the candidate concepts the speaker agent is describing. Both agents receive a shared reward $R$ if the listener correctly identifies the described concept.

$$R(t') = \begin{cases} 1 & \text{if } t = t' \\ 0 & \text{else} \end{cases} \tag{4}$$

The speaker agent consists of a single encoding layer to encode the input vector into a dense representation $h_S$ and an LSTM to produce a message out of $h_S$. The listener agent encodes the message with an embedding layer and an LSTM into a dense representation $h_L$. The listener contains an encoding layer as well, which it applies to every candidate concept respectively to generate a set of representations. It calculates the compliance between message and candidate concepts with the dot product between the message representation and the concept representation. The result is treated as a Gibbs Distribution. Both policies $\pi_S$ and $\pi_L$ output a probability distribution over all possible actions. For the speaker, the possible actions are the elements of the alphabet, once for every symbol over the length of the message. For the listener, the actions consist of choosing each of the candidate concepts in $C$.

For more details see Lazaridou et al. (2018).

**Learning** As part of the reinforcement learning setting, the agents try to maximize the expected reward. They do not share any parameters but try to maximize the probability of their action that resulted in a positive reward, respectively. Therefore together, they maximize the objective function in each training instance:

$$R(t') \left( \sum_{l=1}^{L} \log p_{\pi^S}(m_t^l | m_t^{<l}, h_S) + \log p_{\pi L}(t' | h_L, C) \right) \tag{5}$$

**Empathy Extension** To generate richer gradients for shaping the deep encoding layers of the speaker, we assign an auxiliary unsupervised prediction task to it. We add a single layer Multi-Layer-Perceptron (MLP) to the graph, which predicts the activation of the listener's hidden layer $h_L$ after hearing the full message. The input is the activation of the speaker's hidden layer $h_S$ before starting the utterance. That corresponds to predicting the effect of the to-be-made sentence on the mindset of the listener. We use the mean absolute error as the loss function for the prediction task:

$$loss = \alpha |\sigma(w_\theta(h_S)) - h_L| \tag{6}$$

where $\alpha$ is a weighting factor that ensures that the unsupervised task does not corrupt the main reinforcement learning task. An $\alpha$ close to 1 would mean, that effectively manipulating the listener's mind is as important to the speaker as communicating the target concept. $w_\theta$ is the linear transformation through the MLP with sigmoid activation function $\sigma$. The gradients of the unsupervised task are calculated on the same trial and added to the gradients of the reinforcement task. Hence, no additional training steps are necessary. The optimization then uses the summed up gradient.

We implemented the setup using the EGG toolkit Kharitonov et al..

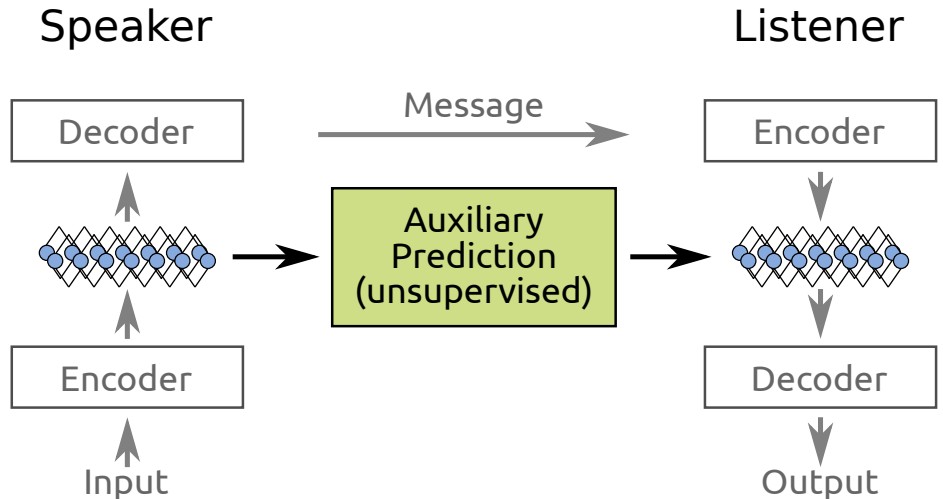

Figure 2: Integration of the auxiliary prediction task of the speaker into the neural network architecture.

## 6 RESULTS

We found that with an $\alpha$ of around 0.1, i.e. weighting the prediction gradients 10% compared to the main task, we can increase the learning speed to double or triple. To be comparable, we used the same initialization and sampling seeds on both options. Good or bad initialization can make up for half the learning speed, but the relative learning speed improvement through using the prediction task stays consistent over different initializations. In Fig. 3 we compared the learning curves with and without the prediction task. For a game setup with two candidate concepts and a maximum message length of two, all marks are reached in half the time. For a more complex game setup with five candidate concepts and a maximum message length of five, some marks are even reached in a third of the time, when using the prediction task.

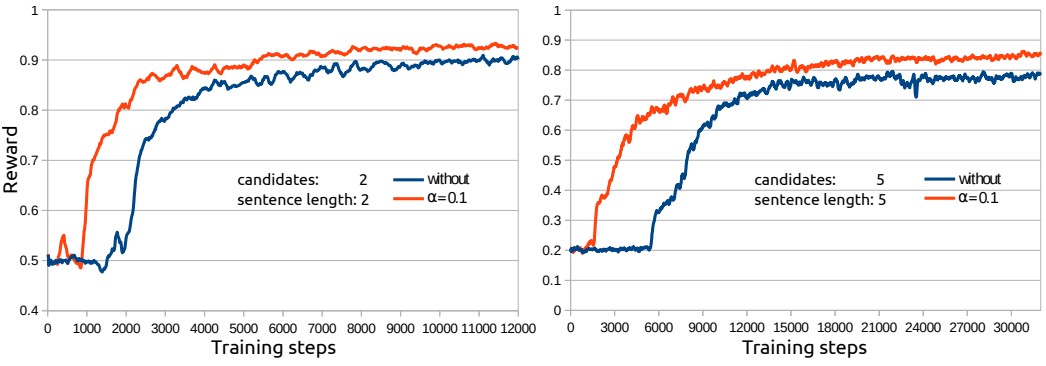

Figure 3: Learning curve for the net with and without prediction task

## 7 CONCLUSION

Using an auxiliary predictive task on a communication learning task has proven auspicious. Sample-efficiency is highly desirable when acquiring language, so the fact that our auxiliary task doubles the learning speed is of high significance. Our experiments do only feature a small partition of the potential of this elegant mechanism, yet. Higher sample-efficiency at no computational cost now allows acquiring more complicated language tasks, that were previously impossible to learn in a reasonable time. We plan to apply our algorithm to much more challenging tasks in the future. We

did, for example, only test disentangled input due to computational limitations. The mechanism would be even more useful when applied on entangled input because developing an expressive representation is then of higher importance. For future research, we propose the use of an auxiliary prediction task for the listener to align with the word usage of the speaker, as well. We hope that this simple but powerful mechanism brings the field of language emergence a big step forward.

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
