# OpenReview forum: "Enhancing Language Emergence through Empathy"
_ICLR.cc/2020/Conference — Reject_

### Official Review · AnonReviewer3 · 2019-10-14
**Official Blind Review #3**

**Rating:** 1

**Review:**

This paper takes the reference-game setup of Lazaridou et al. (2018), as a means of enabling emergent communication, and adds an auxiliary task to demonstrate that this helps with language emergence. The auxiliary task is to enable the speaker to predict the hidden state of the listener, after the message has been received. This is (not unreasonably) likened to providing the speaker with some empathy, in that it enables the speaker to try and predict what the effect of the message will be on the listener.

The main result is that the learning exhibits a speed-up, arriving at roughly the same level of overall reward but in fewer training steps.

The idea of adding an "empathy" auxiliary task to the reference-game setup is an interesting one, and the approach is well-motivated and described, including a background section. Unfortunately, however, the contributions of the paper are some way off what would be required for a full ICLR paper. Note that the main experimental results section takes up only 1/3 of a page, and the overall paper has only 5 pages of content. (As far as I know there is no requirement for an ICLR paper to take up the whole 8 pages, but a submission with only 5 pages is quite unusual.) So the overall contribution could be summarised as taking an existing emergent-language setup with the same speaker and listener neural architectures; adding a single MLP to the speaker; and showing two graphs of training reward, varying the number of candidates (2 and 5). I hope that the authors can perhaps see that this submission would be better suited to a dedicated workshop on emergent communication (and even then it would need more experiments and analysis).


**Experience Assessment:**

I have published in this field for several years.

**Review Assessment: Checking Correctness Of Derivations And Theory:**

I did not assess the derivations or theory.

**Review Assessment: Checking Correctness Of Experiments:**

I assessed the sensibility of the experiments.

**Review Assessment: Thoroughness In Paper Reading:**

I read the paper at least twice and used my best judgement in assessing the paper.

---

### Official Review · AnonReviewer1 · 2019-10-23
**Official Blind Review #1**

**Rating:** 1

**Review:**

This paper starts with a conceptual claim that incorporating a notion of “empathy” in language emergency would help agents learn faster.  The paper then proposes a learning mechanism for implementing this, and looks at its empirical effect for the case of a Speaker-Listener game.

The concept at the core of the paper is thought-provoking, somewhat grounded in human communication, and it’s interesting to see how this can be translated into a learning mechanism for the multi-agent setting.   The specific implementation proposed seems reasonable at a high level, however there are many technical details missing which really hamper the paper’s message & potential scientific impact.   The results are limited to a single game, with just a pairwise comparison (with and without “empathy”), and provide a narrow view into the effectiveness of the proposed technique.

My main problem with the paper is the clarity & organization problems.  Usually I tend to be lenient on this, thinking poor writing is much easier to fix than poor science.  But in this case the problems are large enough that the paper is just too far from the standard for ICLR publication.  It also fits in 5 pages, so the authors had lots of space to write a much better paper.   I encourage them to do this for a future submission, in addition to more extensive results, because I think the ideas are worthwhile.

Specific comments:

Design of the empathy mechanism. Can you motivate why it’s reasonable to “achieve a high relation between the hidden states of both agents”? Is this necessary / sufficient for empathy?  What are alternate framings of this?  What are properties and pros/cons of this framing?

Sec.4 needs a lot more detail!   Sections Agent setup, Learning and Empathy Extension in Sec.5 should be moved to Sec.4, since they describe the method, rather than the experiments.

Sec.5 needs better clarity.
o	What are m^l_t and M^{<l}_t in Eqn 5?  Define how h_S and h_L are parameterized, and how each is trained.
o	Fig.2 gives a high-level view of the approach, but lacks important details.  Do you apply a loss at both the Speaker’s Decoder output, and the Listener’s Decoder output?  Or just the latter?  What is the loss specifically? I assume combination of Eqn 5 and Eqn 6, but not sure.
o	If you train just on the loss of the Listener’s Decoder, does this mean this is backpropagated all the way to train the Speaker?  How would this be done in a real system?  It’s a very strong assumption to say that the Listener will share gradients with the Speaker.  It seems more realistic to assume they will each observe a loss and train independently.

Results are very brief.
o	How robust are the results to the specification of the \alpha (the loss weight from Eqn 6)?  How much data goes to finding a good \alpha?
o	What is the difference between the left and the right plot?  Is one for the Speaker and the other for the Listener?
o	How do you measure “learning speed”, which is the main metric discussed in the text of Sec.6?
o	How do the results change by number of concepts in the game?
o	Do you do any pre-training of the encoder/decoder networks?
o	Can you show confidence intervals on each curve?
o	Can you show test performance?
o	Are there other related games to consider?

Many references missing throughout to support statements, e.g.
o	“Natural language is not as rule-based as…”
o	“These considerations led to the research field of emergent communication…”
o	Sec.2:  Earlier refs to RL in general (e.g. work Sutton in the 1980’s). Earlier refs to RL with neural networks (e.g. work of G. Tesauro; work of M. Riedmiller).
o	Referential game in Fig.1 caption.

Some minor language issues, e.g.
o	“The field was then alleviated” -> Do you mean elevated?


**Experience Assessment:**

I have published one or two papers in this area.

**Review Assessment: Checking Correctness Of Derivations And Theory:**

N/A

**Review Assessment: Checking Correctness Of Experiments:**

I carefully checked the experiments.

**Review Assessment: Thoroughness In Paper Reading:**

I read the paper thoroughly.

---

### Official Review · AnonReviewer2 · 2019-10-24
**Official Blind Review #2**

**Rating:** 1

**Review:**

Summary: This paper aims to take insight from human language acquisition and the importance of empathic connection to learn better models for emergent language. The authors propose an approach to introduce the notion of empathy to multi-agent deep RL by extending existing approaches on referential games with an auxiliary task for the speaker to predict the listener’s empathy/mind. Experiments show that this gives some improvement with faster convergence.

Strengths:
- The concept is interesting and grounded in human communication.

Weaknesses:
- I like the motivation of predicting empathy, but the paper vastly oversells this part: I don't see how predicting the listener's hidden state is the same as modeling empathy. Empathy is a complex human state/emotion that should not be reduced to this.
- This paper is very preliminary. There are multiple typos in the paper. Figures are not professionally created. There are multiple training details not included such as how the agents are modeled and trained. The authors seem to have run out of time given the short length of the paper.
- The experimental results are not convincing at all. The improvement is too small and it would help to run the experiment multiple times to see the improvement with respect to variance in the model. There should also be experiments testing the effect of \alpha on performance. There is no analysis of how well the model is able to predict empathy, as well as ablation studies testing for various design decisions. The authors should also add an analysis of the learned communication protocols and whether they are different in a meaningful way.

**Experience Assessment:**

I have read many papers in this area.

**Review Assessment: Checking Correctness Of Derivations And Theory:**

N/A

**Review Assessment: Checking Correctness Of Experiments:**

I carefully checked the experiments.

**Review Assessment: Thoroughness In Paper Reading:**

I read the paper thoroughly.

---

### Author Response · Authors · 2019-11-11
**Thank you**

I am very grateful for the feedback. It helps me better understand the requirements for a full ICLR paper.
I'm happy, that the reviewers liked the overall idea, I will work on improving the experimental base.

---

### Decision · Program_Chairs · 2019-12-19

**Decision:**

Reject

**Comment:**

This paper introduces the idea of "empathy" to improve learning in communication emergence. The reviewers all agree that the idea is interesting and well described. However, this paper clearly falls short on delivering the detailed and sufficient experiments and results to demonstrate whether and how the idea works.

I thank the authors for submitting this research to ICLR and encourage following up on the reviewers' comments and suggestions for future submission.